# Sarcopenia in Patients with Advanced Gastrointestinal Well-Differentiated Neuroendocrine Tumors

**DOI:** 10.3390/nu16142224

**Published:** 2024-07-11

**Authors:** Elena Romano, Michela Polici, Matteo Marasco, Francesco Lerose, Elisabetta Dell’Unto, Stefano Nardacci, Marta Zerunian, Elsa Iannicelli, Maria Rinzivillo, Andrea Laghi, Bruno Annibale, Francesco Panzuto, Damiano Caruso

**Affiliations:** 1Digestive Disease Unit, Sant’ Andrea University Hospital, ENETS Center of Excellence, 00189 Rome, Italy; elena.romano@uniroma1.it (E.R.); matteo.marasco@uniroma1.it (M.M.); lerose.1802937@studenti.uniroma1.it (F.L.); elisabetta.dellunto@uniroma1.it (E.D.); mrinzivillo@ospedalesantandrea.it (M.R.); bruno.annibale@uniroma1.it (B.A.); 2Radiology Unit, Sant’ Andrea University Hospital, ENETS Center of Excellence, 00189 Rome, Italy; michela.polici@uniroma1.it (M.P.); stefano.nardacci@uniroma1.it (S.N.); marta.zerunian@uniroma1.it (M.Z.); elsa.iannicelli@uniroma1.it (E.I.); andrea.laghi@uniroma1.it (A.L.); damiano.caruso@uniroma1.it (D.C.); 3PhD School in Translational Medicine and Oncology, Department of Medical and Surgical Sciences and Translational Medicine, Faculty of Medicine and Psychology, Sapienza University of Rome, 00185 Rome, Italy; 4Department of Medical Surgical Sciences and Translational Medicine, Sapienza University of Rome, 00189 Rome, Italy

**Keywords:** neuroendocrine neoplasm, somatostatine analog, sarcopenia

## Abstract

Background: Neuroendocrine neoplasms (NENs) are slow-growing tumors. Sarcopenia is defined as the loss of muscle mass, strength, and physical performance. First-line NEN therapy is somatostatin analogs, which could be responsible for malabsorption conditions, such as pancreatic exocrine insufficiency (EPI) with underlying sarcopenia. Aim: Evaluate the prevalence of sarcopenia in patients with NENs at diagnosis and during follow-up. Methods: A retrospective single-center study was conducted, including patients with advanced intestinal NENs G1/G2 (excluded pancreatic NENs). CT scans were analyzed at diagnosis and after 6 months of therapy, and the skeletal muscle index was assessed. Results: A total of 30 patients (F:M = 6:24) were enrolled, with the following primary tumor sites: 25 in the ileum, 1 stomach, 2 jejunum, and 2 duodenum. At diagnosis, 20 patients (66.6%) showed sarcopenic SMI values, and 10 patients (33.3%) showed non-sarcopenic SMI values. At follow-up, three more patients developed sarcopenic SMI values. Statistical significance in relation to the presence of sarcopenia was found in the group of patients with carcinoid syndrome (*p* = 0.0178), EPI (*p* = 0.0018), and weight loss (*p* = 0.0001). Conclusion: Sarcopenia was present in 2/3 of the patients with advanced intestinal NENs at the diagnosis and during the follow-up. It is reasonable to consider this condition to improve clinical outcomes.

## 1. Introduction

Neuroendocrine neoplasms (NENs) represent a heterogeneous group of rare neoplasms originating from the diffuse neuroendocrine cell system, which can arise in various sites, particularly in the gastrointestinal tract, pancreas, and lungs [1]. Despite their rarity, the incidence of NENs is rising, necessitating a deeper understanding of their pathogenesis and management [2]. According to the World Health Organization (WHO) classification, NENs are classified, based on morphology and proliferation index, into well-differentiated neuroendocrine tumors (NETs G1, Ki67 < 3%; NET G2, Ki67 3–20%; NET G3, Ki67 > 20%) and poorly differentiated neuroendocrine carcinomas (NECs, always G3) [3]. This classification plays a significant prognostic role, being the basis for planning treatment and follow-up.

Clinically, NENs can be subdivided into functional or non-functional types; the latter are less frequent and characterized by the secretion of active hormones or bioactive substances, leading to specific clinical syndromes, the most frequent being carcinoid syndrome (characterized by flushing, diarrhea, and bronchospasm with wheezing) [4]. Since most NENs are low-grade and express somatostatin receptors, the first-line therapeutic approach often consists of long-acting somatostatin analogs (SSAs), specifically Lanreotide and Octreotide. SSAs inhibit the release of numerous hormones and suppress cell proliferation and angiogenesis, exerting antiproliferative effects and ameliorating hormonal symptoms [5]. Generally well tolerated, SSAs may cause mostly transient gastrointestinal side effects such as diarrhea, abdominal discomfort, flatulence, and nausea. Long-term use may increase the risk of gallstone formation, cause metabolic disorders such as impaired glucose tolerance, and lead to the malabsorption of vitamins and other nutrients [6]. In particular, it has been shown that SSAs, in approximately 20% of the patients, can lead to exocrine pancreatic insufficiency (EPI), a condition caused by reduced secretion or the inappropriate activity of pancreatic juice and its digestive enzymes, which can cause the malabsorption and maldigestion of lipids and fat-soluble micronutrients, with clinical manifestations such as steatorrhea, weight loss, and asthenia [7,8,9].

The definition of cachexia, initially and for a long time, was focused only on the loss of weight and appetite; recent definitions have attempted to integrate the concept of cachexia as a complex metabolic syndrome, clearly distinct from malnutrition.

In accordance with the literature and international guidelines [10], we can define neoplastic cachexia as a multifactorial syndrome, characterized by the progressive loss of muscle mass (with or without loss of fat mass), which cannot be completely corrected with conventional nutritional support and which leads to progressive functional damage. From a physiopathological point of view, it is characterized by a negative protein and energy balance, mainly caused by metabolic alterations resulting from an underlying chronic inflammatory process which is often associated with reduced caloric intake.

Cachexia is considered different from inanition, the age-related loss of muscle tissue (sarcopenia), nutritional deficits associated with primary depression and malabsorption syndrome, and hyperthyroidism [11]. Furthermore, secondary causes such as the altered integrity and functionality of the gastrointestinal tract, due to mechanical obstruction, intestinal malabsorption from post-surgical syndromes, the toxicity of chemo-radiotherapy treatments, uncontrolled pain, and depression, can contribute to the deterioration of the nutritional status (secondary cachexia).

The characterizing aspects of cancer cachexia are commonly weight loss (especially muscle mass) and inflammation. They are attributable to the main symptoms of the cachectic patient such as anorexia, anemia, and fatigue which contribute to the complex clinical picture and compromise the patient’s quality of life [12]

Recently, the use of standardized methods for assessing body composition, including computed tomography, has facilitated the understanding of the real prevalence of cancer cachexia. Cachexia is a syndromic picture with an incidence of 50–80% in patients suffering from oncological disease and is responsible for the death of at least 20% of them. The etiology is multifactorial and complex, mediated in part by pro-inflammatory cytokines and by specific factors deriving from the tumor which, through acute phase proteins, lead to a significant loss of skeletal muscle even in the presence of adequate food intake and normal insulin levels [13].

The most relevant phenotypic characteristic of cachexia is the loss of muscle mass (sarcopenia) with the appearance or progressive worsening of asthenia, tiredness, the alteration of physical function, the reduction in tolerance to treatments, the compromise of quality of life, and the reduction in survival. Sarcopenia can be present in 20–70% of the cases depending on the type of tumor. Sarcopenia, characterized by low muscle strength, mass, and function, is prevalent among the elderly and can also result from various chronic illnesses, including several types of cancers [1,14]. This condition significantly impacts health because it increases the risk of falls, fractures, and hospitalizations. Moreover, it impairs the ability to perform the activities of daily living, leading to a decline in the overall quality of life. According to the EWGSOP2 2018 definition, sarcopenia is defined as probable when low muscle strength is detected and as confirmed in the presence of low muscle quantity or quality. When low muscle strength, low muscle quantity/quality, and low physical performance are detected, sarcopenia is considered severe.

A variety of tests and tools are now available for the detection of sarcopenia in clinical practice and research, such as (A) SARC-F, a five-item questionnaire self-reported by patients as a screen for sarcopenia risk; (B) the grip strength measurement with a dynamometer; (C) imaging techniques such as Magnetic Resonance Imaging (MRI) and computed tomography (CT), which are considered gold standards for the non-invasive assessment of muscle quantity/mass. In particular, the CT images of a specific lumbar vertebral landmark (L3) correlate significantly with whole-body muscle mass, so this imaging method has been used to detect low muscle mass and predict prognosis [15]. Sarcopenia must be differentiated from cancer cachexia. Following the definition proposed by Fearon in an international consensus [16], cancer cachexia is a multifactorial syndrome characterized by an ongoing loss of skeletal muscle mass (with or without the loss of fat mass) that cannot be entirely reversed by conventional nutritional support. Patients who have more than 5% loss of stable body weight over the past 6 months (in the absence of simple starvation), or a body mass index (BMI) less than 20 kg/m^2^ and an ongoing weight loss of more than 2%, or sarcopenia and an ongoing weight loss of more than 2%, are classified as having cachexia. 

Cancer cachexia is a spectrum in which three stages can be recognized (precachexia, cachexia, and refractory cachexia), even if not all the patients traverse all three stages.

Sarcopenia, a crucial component for the diagnosis of cachexia, is mainly overlooked and untreated in general clinical practice, leading to high personal, social, and economic burdens. While a direct link between NENs and sarcopenia remains elusive, emerging evidence suggests a potential role of sarcopenia in NEN patients, which might be increased by the metabolic and hormonal disturbances induced by SSAs in these patients [17,18]. However, few data are available regarding the prevalence of sarcopenia in patients with gastrointestinal NENs and its potential prognostic role in this clinical context [19,20].

This study aimed to evaluate the prevalence of sarcopenia in a group of patients affected by advanced gastrointestinal well-differentiated G1-G2 NETs at the time of diagnosis and during follow-up.

## 2. Materials and Methods

### 2.1. Study Design

This single-center retrospective study was conducted at the Sant ‘Andrea University Hospital, ENETS Center of Excellence of Rome. The study enrolled consecutive patients evaluated at the center between 2019 and 2023 who met the following inclusion criteria: i. the histologically proven diagnosis of NET G1 or NET G2; ii. the gastrointestinal origin of the primary tumor; iii. advanced/unresectable disease, candidates for treatment with SSAs; iv. the availability of CT images taken within 6 months of starting SSA therapy; v. the availability of CT images taken during follow-up; vi. available data on fecal elastase 1 at the time of diagnosis.

The patients with an indeterminate primary tumor location were considered to be part of the small bowel group if they exhibited symptoms of carcinoid syndrome and met specific histological standards, once other likely primary sites were excluded using standard imaging techniques (CT or MRI, and 68Ga-PET), as previously described [21].

Gastrointestinal NENs were categorized according to the WHO 2019 classification and TNM staging system. The expression of somatostatin receptors was assessed by 68Gallium-radiolabeled Positron Emission Tomography (68Ga-DOTA-PET). 

To reduce the heterogeneity of the patient population, the patients with NET G3 were excluded, as well as the patients with pancreatic NENs, in order to avoid bias related to possible chronic obstructive pancreatic damage, which might have altered pancreatic function with potential consequences on nutrient digestion [22]. Carcinoid syndrome was diagnosed in the presence of specific symptoms and elevated 5-HIAA levels following the ENETS recommendations [4].

### 2.2. Evaluation of Sarcopenia

Sarcopenia was assessed by evaluating unenhanced CT scans at the NET diagnosis and during the first follow-up CT (performed according to the ENETS guidelines, 3–6 months after the beginning of the SSA treatment). Skeletal muscle mass was assessed by a semi-automatic segmentation method using dedicated software (NIH ImageJ 1.46) [23]. An axial image at the level of the third lumbar vertebra L3 was used to calculate the skeletal muscle mass using specific attenuation thresholds (−29, +150) to assess the skeletal muscle mass [24]. Specific regions of interest (ROIs) were used to calculate the outer and inner musculature perimeter of the skeletal muscle mass and the vertebral body perimeter (Figure 1).

The skeletal muscle mass (SMM) was calculated by subtracting the area of the internal musculature and the vertebral body from the area of the external musculature and then dividing the result by 100.
$$\mathrm{SMM}=\left(\mathrm{Outert}\;\mathrm{Abdominal}\;\mathrm{Muscle}\;\mathrm{area}-\mathrm{Inner}\;\mathrm{Abdominal}\;\mathrm{Muscle}\;\mathrm{area}-\;\mathrm{Vertebral}\;\mathrm{body}\;\mathrm{area}\right)/100$$

The skeletal muscle index (SMI, cm^2^/m^2^), necessary for assessing sarcopenic status, was finally calculated by normalizing the MS by the patient’s height squared [25].
$$\mathrm{SMI}=\mathrm{SM}/h^{2}$$

The SMI cut-offs for sarcopenia are SMI < 52.4 cm^2^/m^2^ in men and SMI < 38.5 cm^2^/m^2^ in women [17].

### 2.3. Statistical Analysis

The quantitative data were expressed as medians, ranges, absolute numbers, and percentages. The variables were analyzed using Fisher’s exact, chi-squared, and Mann–Whitney tests. The differences were considered statistically significant when the *p*-value was <0.05. Statistical analysis was performed using dedicated software (MedCalc v.21) (https://www.medcalc.org/). The relevant ethics committees approved the study, adhering to local legislation, and informed consent for data collection was obtained from all the patients (Ref 5580).

## 3. Results

A total of 30 patients with gastrointestinal G1 and G2 NET were enrolled, comprising six females and 24 males. The patients’ general features are summarized in Table 1. 

The majority of the patients (*n* = 25, 83.3%) had their primary tumor located in the ileum, while the remaining five patients (16.6%) had primary tumors in other gastrointestinal sites (duodenum, *n* = 2; jejunum, *n* = 2; stomach, *n* = 1). The median age at the NET diagnosis was 60 years (range 40–81).

At the time of the initial diagnosis, 14 patients (46.7%) had metastatic disease. Regarding tumor grading, 20 patients (66.6%) were classified as having NET G1, while the remaining 10 (33.3%) had NET G2. The median Ki-67 proliferative index was 2% (range 1–10%).

Regarding functional status, ten patients (33.3%) had tumor-related carcinoid syndrome and were classified as having ‘functioning tumors’. At the time of the diagnosis, before any medical treatment was initiated, five patients (16.6%) were diagnosed with EPI with FE-1 <200 mcg/g stool, and eight patients (26.6%) had diabetes. The primary tumor had been surgically resected in 20 patients (66.6%).

All the patients diagnosed with metastatic NETs underwent SSA therapy. Specifically, 16 patients (53.3%) received Lanreotide autogel at a dose of 120 mg/4 weeks, and 14 patients (46.6%) were treated with Octreotide LAR at a dose of 30 mg/4 weeks.

When the baseline CT images were evaluated to assess sarcopenia, the mean SMI at diagnosis was 46.18 ± 7.87 cm^2^/m^2^ for men and 44.82 ± 7.77 cm^2^/m^2^ for women, as shown in Figure 2. Based on this figure, 20 patients (66.6%) exhibited SMI values compatible with sarcopenia, while 10 patients (33.3%) presented non-sarcopenic SMI values.

A statistically significant correlation was observed between sarcopenia and the presence of carcinoid syndrome (*p* = 0.0178), co-existent EPI (*p* = 0.0018), and the presence of weight loss (*p* = 0.0001). At the time of the first follow-up CT (6 months after the initiation of the SSA therapy), three additional patients (10%) showed SMI values compatible with sarcopenia, ten patients (33.3%) experienced a worsening of pre-existing sarcopenia, and ten patients (33.3%) remained in a stable, persistent sarcopenic condition. Consequently, six months after starting the SSA treatment, 23 patients (76.7%) were found to be sarcopenic, and 7 patients (23.3%) were non-sarcopenic (Table 2).

## 4. Discussion

Neuroendocrine gastrointestinal neoplasms (NENs) are clinically diverse and biologically complex diseases that pose significant diagnostic and management challenges. Emerging evidence suggests a potential correlation between neuroendocrine tumors, SSA therapy, and malnutrition, raising concerns about treatment-related complications and therapeutic management strategies.

SSA therapy, through binding to the somatostatin receptors expressed on the cell surface of most NENs, has a dual role in symptom control and tumor growth inhibition, as well as a favorable safety profile. However, since their side-effects are predominantly gastrointestinal (diarrhea, nausea, and the malabsorption of nutrients and fats), SSAs can have a negative impact on the nutritional status of patients. [6].

Clement et al. [26] described the potential role of somatostatin analog on nutritional status in patients with gastroenteropancreatic neuroendocrine tumors. In this study, the prevalence of malnutrition based on GLIM criteria [27] was 70% in patients with GEP-NETs using SSA, in line with our results.

The possible role of somatostatin analog, as previously explained, could be correlated with pancreatic exocrine insufficiency and reduced nutritional uptake contributing to sarcopenia.

Nutritional status could be related to the excessive production of gastrointestinal hormones, peptides, and amines, which can lead to malabsorption, diarrhea, steatorrhea, and altered gastrointestinal motility; in particular, the medical management of NETs can lead to the alteration of gastrointestinal secretory, motor, and absorptive functions, with both dietary and nutritional consequences [28].

Sarcopenia, as reported by Shafiee [29] in a meta-analysis of thirty-five general population studies, has an overall prevalence of 10%, is traditionally associated with aging, and is increasingly recognized as prevalent and debilitating in patients with various tumors, including NENs, exerting detrimental effects on treatment outcomes and the quality of life [14].

Cancer-related sarcopenia not only diminishes the quality of life but also contributes to treatment-related toxicity, impaired physical function, and inferior outcomes, potentially leading to cachexia, a condition that, as outlined previously, is only partially reversible with the standard nutritional interventions. For these reasons, cancer-related sarcopenia poses a significant public health concern, necessitating early detection and intervention strategies. Despite the established link between sarcopenia and several non-neuroendocrine cancers, few studies have explored the prevalence of sarcopenia in patients with NENs. In a notable study by Clement [19], sarcopenia was present in 69% of the population with gastrointestinal and pancreatic NENs. This prevalence is notably higher compared to 27–44% in patients with metastatic gastrointestinal non-neuroendocrine cancers [30,31,32]. A recent review [18] emphasized the role of sarcopenia in NEN patients, though most studies reviewed involved neuroendocrine carcinoma, known for its aggressive behavior similar to adenocarcinoma.

Regarding assessment modalities, the limited literature has investigated the role of anthropometric data and BMI as the predictors of sarcopenia [18,19,20]. It is well known that relying solely on BMI to evaluate nutritional status is incorrect, as it risks misdiagnosing malnutrition, which may be present in patients with cancer who have sarcopenic obesity [33]. This finding is confirmed in our study of gastrointestinal NEN patients, where normal BMI values were recorded despite a significant proportion of sarcopenic patients. This finding underscores the inaccuracy of using anthropometric data alone to assess sarcopenia in NEN patients. The quantitative non-invasive assessment of muscle mass is crucial for identifying patients at risk of complications related to sarcopenia. Imaging techniques such as CT scans are widely available and valuable as they are already the preferred method for tumor staging and monitoring during treatment and follow-up [34]. This makes CT an excellent tool for evaluating muscle mass in addition to its use in oncological assessments.

It is noteworthy that the high prevalence of sarcopenia reported in this study pertains to a selected population of gastrointestinal well-differentiated NETs. While the risk of sarcopenia in pancreatic NENs was reported to be comparable to that of pancreatic adenocarcinoma (up to 86%), specific data on non-pancreatic NENs are scarce [32,35,36]. The high proportion of sarcopenia in patients with tumors not involving the pancreatic gland suggests that factors other than the primary tumor site contribute to this condition.

Interestingly, a recent study observed that some patients developed sarcopenia within months of starting SSA treatment. SSAs may worsen malnutrition and muscle wasting by inhibiting exocrine pancreatic secretion, impairing nutrient absorption, and inducing hormonal and metabolic changes that promote muscle catabolism and hinder muscle regeneration [37,38]. Consequently, SSA therapy, mainly by inducing EPI, could exacerbate pre-existing sub-optimal nutritional conditions. This finding underscores the importance of thorough nutritional assessments for NEN patients, similar to recommendations for other cancers. It has been estimated that about 50% of cancer patients already had pre-existing sarcopenia before starting tumor-related treatments [39]. Malnutrition and low muscle mass critically affect treatment tolerance, quality of life, and survival, making it imperative to screen and assess all cancer patients for malnutrition routinely. Multimodal interventions, including nutritional support, exercise training, and supportive care, are vital to mitigate muscle loss and enhance functional capacity.

This study, while providing valuable insights into sarcopenia in patients with gastrointestinal NENs, is subject to several limitations. The most significant of these is the small sample size, which may limit the generalizability of our findings and which could explain the higher prevalence of sarcopenia compared to that of the general population [26]. Additionally, the retrospective design of this study restricts our ability to draw definitive causal inferences between treatment with SSAs and the progression of sarcopenia. Furthermore, the absence of long-term follow-up data hampers our understanding of the evolution of sarcopenia over time and its impact on clinical outcomes for these patients.

Despite these limitations, the study benefits from a highly homogeneous patient population consisting solely of individuals with well-differentiated G1 and G2 gastrointestinal NETs. This homogeneity enhances the consistency of the data and the validity of our findings within this specific group. Moreover, the rarity of this disease underscores the importance and relevance of our study, as data on this patient population are scarce, making even preliminary insights extremely valuable for the medical community.

## 5. Conclusions

This study shows that about two-thirds of these individuals exhibit sarcopenia when they are first diagnosed, a condition that might worsen during treatment. We strongly recommend routine non-invasive evaluations for sarcopenia at both diagnosis and during follow-up in patients with gastrointestinal NENs, particularly those being considered for SSA treatment. 

We also suggest that sarcopenia in patients with NENs should be managed in a multidisciplinary way, including nutritional and physical interventions, at the moment of diagnosis and during all the therapeutic strategy periods, with the help of dedicated professional figures such as dietitians and physical therapists. Collaborative care ensures that all the aspects of the patient’s health are addressed comprehensively.

It would be important to provide a tailored diet with a correct caloric and protein intake to meet the increased metabolic demands of cancer patients and prevent muscle wasting. For patients struggling to meet their nutritional needs through diet alone, nutritional supplements could be beneficial. Moreover, strength training exercises, tailored to the patient’s capabilities, should be suggested to help stimulate muscle hypertrophy and improve muscle strength. Finally, educating patients and their caregivers about the importance of nutrition and physical activity is crucial.

These results highlight the importance of a constant morphological and nutritional structure in cancer patients at diagnosis and during ‘disease-oriented’ therapies. The study also underlines how sarcopenia is a real problem also in the oncology population, too often not evaluated or misinterpreted in the context of ‘cluster symptoms’ and, therefore, worthy of clinical studies dedicated to expanding knowledge and developing adequate diagnosis and prevention strategies for the phenomenon.

To improve patient care and results, it is essential to have a comprehensive grasp of the pathophysiology, diagnostic methods, and treatment options specifically for sarcopenia related to NEN. Future studies should concentrate on longitudinal CT scan evaluations to better determine the effects of SSA treatment and other therapies on the progression of sarcopenia in patients with NEN.

## Figures and Tables

**Figure 1 nutrients-16-02224-f001:**
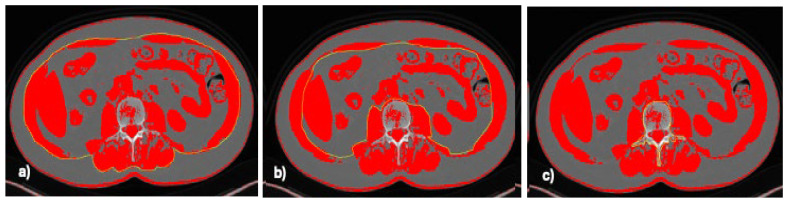
Evaluation of skeletal muscle index in CT Scan in patients with ileal NET. (**a**) Outer muscle perimeter of skeletal muscle mass (yellow line); (**b**) inner muscle perimeter of skeletal muscle mass (yellow line); (**c**) vertebral body perimeter of skeletal muscle mass (yellow line).

**Figure 2 nutrients-16-02224-f002:**
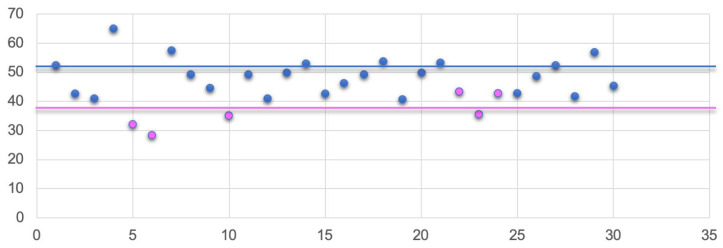
Distribution of SMI values (*n* = 30 patients) at the time of diagnosis. Females’ values are in pink, and males’ values are in blue. Continuous horizontal lines refer to SMI cut-off values for sarcopenia.

**Table 1 nutrients-16-02224-t001:** Patients’ general characteristics.

Patients’ General Characteristics	N (%)
Sex	
Male	24 (80%)
Female	6 (20%)
Median age at diagnosis (range)	62 (40–81%)
Primary tumor site	
Ileum	25 (83.3%)
Stomach	1 (3.3%)
Duodenum	2 (6.6%)
Jejunum	2 (6.6%)
Metastatic disease at diagnosis (M1)	14 (46.6%)
Surgery before therapy with SSAs	20 (66.6%)
Median Ki67 value	2 (1–10%)
Grade	
G1	20 (66.6%)
G2	10 (33.3%)
Carcinoid syndrome	10 (33.3%)
Pancreatic exocrine insufficiency	5 (16.6%)
Diabetes	8 (26.6%)
BMI	
<18 (underweight)	0 (0%)
18–25 (normal weight)	12 (40%)
25–30 (overweight)	13 (43.3%)
>30 (obesity)	5 (16.7%)

**Table 2 nutrients-16-02224-t002:** Comparison between patients with or without sarcopenia.

	NENs with Sarcopenia at Diagnosis (%)	NENs without Sarcopenia at Diagnosis (%)	*p* Value
EPI	4 (13%)	1 (3%)	0.0018
Carcinoid syndrome	8 (26.6%)	2 (6.6%)	0.0178
Weight Loss	2 (6.6%)	1 (3%)	0.0001
Male vs. Female	16 (53.3%) vs. 4 (13%)	8 (26.6%) vs. 2 (6.6%)	Not significant
G1 vs. G2	14 (46%) vs. 6 (20%)	6 (20%) vs. 4 (13%)	Not significant

## Data Availability

The data presented in this study are not readily available due to privacy restrictions. Requests to access the datasets should be directed to the corresponding author.

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
