# Peer review of "Sarcopenia in Patients with Advanced Gastrointestinal Well-Differentiated Neuroendocrine Tumors"

_nutrients, 2024, doi:10.3390/nu16142224_

Round 1
Reviewer 1 Report
Comments and Suggestions for Authors
Comment
This manuscript clinically reported on the skeletal muscle index (SMI) in patients with neuroendocrine neoplasms (NENs). It is relatively unique as an article in Nutrients. There are two results: One is Figure 2, which probably shows no relationship (without statistical analysis) between SMI and the time of diagnosis. The other is Table 2, which shows that NENs with sarcopenia-like SMI have exocrine pancreatic insufficiency (EPI), carcinoid syndrome, and weight loss.
I’m wondering what the difference is between “sarcopenia with NENs” and “cancer cachexia”. The median age of the patients in this manuscript is 62 years (Table 1), which is not as old as sarcopenia patients. The prevalence of sarcopenia in people older than 60 years is 10% (Shafiee et al., Journal of Diabetes and Metabolic Disorders, 2017; 16: 21). On the other hand, two-thirds of the patients in this study had a sarcopenia-like SMI, which is too high as a ratio of sarcopenia. There are some studies and reviews describing cachexia in NEN patients (Poblocki et al., Nutrients, 2020; 12: 1437). Please describe, explain, and/or discuss why the authors did not mention cachexia at all in the manuscript.
For publication in Nutrients, the manuscript needs to be fundamentally restructured with the definition of sarcopenia and cachexia and the precise description of the physiological parameters based on the definition.
Minor comments
1. Materials and Methods should be described separately with subtitles.
Author Response
Thank you for the review. I agree with his assessment, and we have integrated the work as suggested. Please find attached a point by point response to your comments.

Reviewer 2 Report
Comments and Suggestions for Authors
- The manuscript provides a concise overview of the study on the prevalence of sarcopenia in patients with neuroendocrine neoplasms (NENs) at diagnosis and during follow-up. The study's aim, methods, results, and conclusions are clearly outlined, but there are areas that could benefit from further refinement and clarity.
- The manuscript would benefit from smoother transitions between sentences and a more logical flow of information. For instance, explaining the rationale behind using somatostatin analogues before discussing their potential impact on sarcopenia could enhance clarity.
- More details on the inclusion and exclusion criteria would improve the transparency of the study design. Specifically, why pancreatic NENs were excluded and the rationale behind selecting only advanced intestinal NENs G1/G2 should be briefly mentioned.
- The manuscript mentions that 20 patients showed sarcopenic SMI values at diagnosis, but it would be useful to provide percentages for better context. Also, the follow-up period duration should be specified.
- The conclusion states that sarcopenia was present in 2/3 of patients, which is an important finding. However, elaborating on how this condition can be managed or suggesting specific clinical interventions would strengthen the practical implications of the study.
- The term "indolent" used to describe NENs should be defined or replaced with "slow-growing" for clarity, considering the diverse audience.
The manuscript presents a significant study on the prevalence of sarcopenia in patients with advanced intestinal NENs. It successfully highlights the clinical relevance of recognizing sarcopenia. However, improvements in clarity, detail on methods, statistical analysis, and practical implications are needed to enhance the overall quality and impact of the manuscript. Addressing typographical issues and providing more demographic and clinical context would also improve the reader's understanding and engagement with the study findings.
Author Response

(The authors gave the same response as above.)

Round 2
Reviewer 1 Report
Comments and Suggestions for Authors
The authors revised the manuscript according to the reviewer's comments.